# DNA G-Quadruplexes Contribute to CTCF Recruitment

**DOI:** 10.3390/ijms22137090

**Published:** 2021-06-30

**Authors:** Polina Tikhonova, Iulia Pavlova, Ekaterina Isaakova, Vladimir Tsvetkov, Alexandra Bogomazova, Tatjana Vedekhina, Artem V. Luzhin, Rinat Sultanov, Vjacheslav Severov, Ksenia Klimina, Omar L. Kantidze, Galina Pozmogova, Maria Lagarkova, Anna Varizhuk

**Affiliations:** 1Federal Research and Clinical Center of Physical-Chemical Medicine of Federal Medical Biological Agency, 119435 Moscow, Russia; tikhonova.polly@mail.ru (P.T.); pavlova.yui@phystech.edu (I.P.); isaakova.kate@gmail.com (E.I.); abogomazova@gmail.com (A.B.); neglect1@yandex.ru (T.V.); rhenium112@gmail.com (R.S.); wws83@yandex.ru (V.S.); ppp843@yandex.ru (K.K.); pozmge@gmail.com (G.P.); 2Moscow Institute of Physics and Technology, 141701 Dolgoprudny, Russia; 3A.V. Topchiev Institute of Petrochemical Synthesis RAS, 119071 Moscow, Russia; 4WCRC Russia “Digital Biodesign and Personalized Healthcare”, Sechenov First Moscow State Medical University, 119146 Moscow, Russia; 5Center for Precision Genome Editing and Genetic Technologies for Biomedicine, Federal Research and Clinical Center of Physical-Chemical Medicine of Federal Medical Biological Agency, 119435 Moscow, Russia; 6Institute of Gene Biology Russian Academy of Sciences, 119334 Moscow, Russia; artyom.luzhin@gmail.com (A.V.L.); o_kantidze@mail.ru (O.L.K.)

**Keywords:** G-quadruplex, chromatin remodeling, CpG methylation, CTCF, HMG proteins

## Abstract

G-quadruplex (G4) sites in the human genome frequently colocalize with CCCTC-binding factor (CTCF)-bound sites in CpG islands (CGIs). We aimed to clarify the role of G4s in CTCF positioning. Molecular modeling data suggested direct interactions, so we performed in vitro binding assays with quadruplex-forming sequences from CGIs in the human genome. G4s bound CTCF with Kd values similar to that of the control duplex, while respective i-motifs exhibited no affinity for CTCF. Using ChIP-qPCR assays, we showed that G4-stabilizing ligands enhance CTCF occupancy at a G4-prone site in STAT3 gene. In view of the reportedly increased CTCF affinity for hypomethylated DNA, we next questioned whether G4s also facilitate CTCF recruitment to CGIs via protecting CpG sites from methylation. Bioinformatics analysis of previously published data argued against such a possibility. Finally, we questioned whether G4s facilitate CTCF recruitment by affecting chromatin structure. We showed that three architectural chromatin proteins of the high mobility group colocalize with G4s in the genome and recognize parallel-stranded or mixed-topology G4s in vitro. One of such proteins, HMGN3, contributes to the association between G4s and CTCF according to our bioinformatics analysis. These findings support both direct and indirect roles of G4s in CTCF recruitment.

## 1. Introduction

The guanine quadruplex (G4) structures play a role in DNA replication, transcription, recombination, and DNA repair [1]. G4s may also contribute to epigenetic regulation [2,3] by shaping the DNA methylome [4] and by promoting Yin Yang 1-mediated [5] or CCCTC-binding factor (CTCF)-mediated [6] chromatin looping. G4-chromatin immunoprecipitation sequencing (G4-ChIP-seq) peaks show statistically significant colocalization with CTCF-bound sites [6]. However, the role of G4s in CTCF recruitment has not been explained so far.

Similarly to other zinc-finger (ZF) transcription factors, CTCF binds DNA in a sequence-specific manner, but its cognate sites are rather diverse [7]. A motif that best distinguishes CTCF binding sites from their flanking regions is a 20-mer with a 14bp core CCGCGNGGNGGCAG [8]. A different consensus motif, NCANNAG(G/A)NGGC(G/A)(C/G)(T/C), has been revealed by chromatin immunoprecipitation with exonuclease digestion [9]. It is enriched in most CTCF ChIP-seq peaks and partially matches the five-triplet sequence predicted based on the DNA binding specificity of CTCF ZF3-7 (CCAGCAGGGGGCGCT) [10]. None of these sequences match the consensus G4 motif G_3+_NG_3+_NG_3+_NG_3+_, but all of them are G-rich and may overlap G4-prone sites. To clarify whether G4s can be recognized by CTCF, in vitro binding assays are needed.

CTCF sensitivity to DNA methylation is an open question. Hypomethylation was initially shown to promote CTCF binding [11]; but subsequent whole-genome analyses argued against this possibility [12]. Within CTCF binding motifs, methylation effects on the binding efficiency can be negative or positive depending on the position of the methylated CpG [10]. Interestingly, CTCF is frequently recruited to CpG islands (CGIs) [13], which are typically hypomethylated. Within CGIs CTCF binding is mostly invariant and predetermined by sequence [14]. G4s are enriched in CGIs and facilitate their hypomethylation maintenance [4]; thus, they may explain the CGI-CTCF link. This assumption could be verified using bioinformatic approaches.

Methylation is not the only epigenetic factor at play in CTCF positioning. Moderate nucleosome density and regular nucleosome positioning [15,16] are required for CTCF to access to its cognate sites in linker (internucleosomal) DNA. The relation between chromatin compactness and CTCF has not been analyzed thoroughly so far. It is reasonable to assume that internucleosome contacts would hamper CTCF binding to linker DNA similarly to irregular nucleosome positioning and high nucleosome density. Importantly, nucleosome arrangement is controlled by CpG methylation-sensitive histone modifiers and other chromatin remodelers [15,16]. CTCF is also a chromatin remodeler, capable of nucleosome repositioning [17]. Thus, CTCF recruitment to DNA is regulated via several interlinked pathways (Figure 1), and G4s may contribute to each of them. G4s affect methylation via sequestering DNA methyltransferase DNMT1 from its target sites [4] and may promote chromatin remodeling due to their affinity for architectural chromatin proteins and modifiers, including polycomb related complex 2 subunits [18,19].

This study was designed to partially verify the links between G4s, CGIs, chromatin compactness and CTCF recruitment. First, we used ChIP-seq, in silico modeling and in vitro binding assays to test CTCF affinity for G4s. Next, we analyzed previously reported ChIP-seq and bisulfite sequencing data to determine whether G4s account for CTCF recruitment to CGIs and whether DNMT1 plays a substantial role. Finally, to determine whether G4s affect accessibility of CTCF binding sites in internucleosomal DNA, we explored G4 interactions with linker histone analogs and other architectural chromatin proteins.

## 2. Results

### 2.1. G4s Colocalize with CTCF-Bound Sites in the Genome and Interact with CTCF In Vitro

CTCF is presumed to be incapable of recognizing G4s [2]. However, evidence against such interactions is limited. In exon 1 of the human telomerase reverse transcriptase (hTERT) gene, G4 formation disrupts CTCF binding, resulting in transcriptional repression [20]. Notably, the CTCF binding site in the hTERT gene is unusual in that it adopts either a hairpin or a G4 structure in vitro depending on the CpG methylation status. The G4/hairpin competition may be an hTERT-specific case rather than a representative example. To get a broader picture, we analyzed genome-wide CTCF occupancy under G4-favoring conditions. For that, we treated K562 cells with a G4 stabilizer pyridostatin (PDS) [21] at a concentration of 10 μM for 24 h and performed ChIP-seq experiments.

The resulting CTCF occupancy profile was similar to that obtained for non-treated (control) cells (Figure 2A,B). Both profiles agreed with the previously reported ChIP-seq data (Figure 2C) [6]. CTCF peaks frequently colocalized with or were flanked by G4 motifs, G4-seq-peaks, and C4-ChIP-seq (BG4) peaks (Figure 2D–F). Importantly, CTCF-bound sites in PDS-treated cells intersected with G4-seq peaks more frequently than in non-treated cells, although colocalization was significant in both cases (Fisher exact test OR = 3.6, *p*-value < 1 × 10^−300^ and OR = 3.2, *p*-value < 1 × 10^−300^ for PDS-treated and control cells, respectively). In contrast, BG4 peaks were clearly enriched in CTCF-bound sites irrespectively of PDS (Figure 2D).

We next questioned whether prolonged incubation with an excess of G4-stabilizing ligands would enhance CTCF occupancy at G4 sites. For that, we treated K562 cells with 20 μM PDS for 48 h. Phen-DC3, a more potent G4 stabilizer than PDS [22], was tested in parallel at a concentration of 8 μM. Both PDS and Phen-DC3 are presumed to be pan-quadruplex stabilizers [22,23]. However, their effects on conformationally polymorphic G4-prone sites, such as MYC NHE [24], require further investigation. Changes in CTCF occupancy at a G4-prone site (STAT3), G4 clusters colocalized with (BG4) peaks (MYC and VEGFA), and non-G4 sites (USP24 and VPS4A) were evaluated using ChiP-qPCR (the sites are marked with black arrows in Figure 3A; see Appendix A for primer sequences). The ligands increased CTCF occupancy at the G4-prone site, had minor effects on the BG4-positive sites, and did not affect the non-G4 sites (Figure 3B). These data argued against the common assumption that G4 folding attenuates CTCF binding [2], which prompted us to further explore the possibility of G4-CTCF complexes.

We first investigated direct CTCF-G4 interactions in silico. The well-characterized G4 from MYC NHE, previously referred to as Pu27 G4 [25], was selected as a model structure. Docking of MYC-G4 to CTCF yielded a complex that was similar to a previously reported CTCF-duplex complex [10] (Figure 4A). The G4 fit between CTCF ZF4–ZF7, which ‘embraced’ the G/C-rich B-DNA in reported crystal structures [10], and contacted the protein surface with maximal electrostatic potential. Additional details on the best binding energy conformation of MYC-G4-CTCF and H-bonding in the complex can be found in Appendix A. Briefly, modeling data supported G4 recognition by CTCF.

To verify affinity for CTCF in vitro, we picked three sequences predicted to form stable G4s (red arrows in Figure 3A; Appendix A) from MYC, BDNF, and SHANK1 and performed microscale thermophoresis (MST)-based binding assays with fluorophore-labeled CTCF. Respective i-motifs (Appendix A) were tested in parallel with the G4s, and the consensus CTCF-binding duplex with a core sequence predicted based on the CTCF ZF3-7 specificity [10] was used as a positive control. Secondary structures of all G4 and i-motif oligonucleotides (ODNs) were confirmed by the characteristic circular dichroism (CD) signatures, i.e., peaks at 265/295 and 288 nm, respectively (Figure 4B). All G4s bound CTCF rather efficiently, with Kd values of 140 ± 60 nM (MYC), 120 ± 30 nM (BDN), and 70 ± 30 nM (SHA). Their affinity was comparable to that of a control duplex (Kd = 80 ± 10 nM), while i-motifs showed no binding (Figure 4C).

This finding agrees with G4 and i-motif distribution in the genome relative to CTCF-bound sites (Figure 4D). Approximately 47% of G4 motifs found within BG4 peaks and only 8% of i-motifs predicted to withstand physiologic conditions [26] colocalized with CTCF-bound sites in K562 cells. We conclude that G4s form complexes with CTCF *in vitro* and are associated with CTCF in the genome.

### 2.2. G4s Account for CTCF Occupancy in CGIs Irrespectively of DNMT1 Inhibition

CTCF binding is CpG methylation-dependent [10] and frequently observed in CGIs [6]. The maintenance of CGI hypomethylation is attributed to DNMT1 inhibition by G4s [4]. Therefore, G4s appear to facilitate CTCF binding to CGI via the DNMT1-dependent mechanism (Figure 5A). We performed statistical analysis of the available data to verify this hypothesis.

Predominant localization of G4s in CGIs, CGI hypomethylation, and partial but significant colocalization between G4s and DNMT1-bound sites have been previously demonstrated [4]. In the K562 cell line, 77% of G4 sites identified by G4-ChIP-seq using the BG4 antibody (representing high-confidence BG4 peaks) overlapped with CGIs, and half intersected with DNMT1-bound sites (Figure 5A, Venn diagram). We investigated distribution of these sites relative to CTCF ChIP-seq peaks and then used X^2^ statistics to elucidate whether G4s account for CTCF binding in CGIs.

The frequency of colocalization with CTCF peaks was similar for BG4 peaks inside (63%, 4333/6882) and those outside (62%, 1277/2073) CGIs. On the other hand, BG4-harboring and BG4-lacking CGIs showed markedly different distributions. In most cases (62%, 13249/21399) the latter were CTCF-free, while 9% (1962/21399) flanked and 29% (6188/21399) overlapped with CTCF peaks. In contrast, most BG4-harboring CGIs (72%, 4562/6319) overlapped with CTCF peaks, while 7% (439/6319) flanked CTCF peaks and 21% (1318/6319) were CTCF-free. To summarize, G4s colocalized with CTCF-bound sites irrespectively of CGI presence, while CGIs colocalized with CTCF if they contained G4s (Figure 5B). These results indicate that G4s significantly contribute to the association between CGIs and CTCF (X^2^[1, N = 27,718] = 3848.5, *p* < 0.01) but not vice versa.

We next assessed the contribution of DNMT1. Overlap with CTCF peaks was observed for 62% (858/1385) of DNMT1-lacking BG4-harboring CGIs and 75% (3704/4934) of DNMT1- and BG4-harboring CGIs. Analogously, DNMT1-harboring BG4 peaks within CGIs overlapped CTCF peaks at a slightly higher frequency (68%, 2305/3400) than those lacking DNMT1 (58%, 2028/3482). To summarize, DNMT1-bound sites colocalized with CTCF-bound sites only slightly more frequently than DNMT1-free sites (Figure 5B). This result argues against the decisive role of DNMT1 in CTCF recruitment to CGIs.

Finally, to clarify the role of CpG methylation, we compared methylation levels within CTCF-bound sites, CGIs, and G4 sites in K562 cells using available whole-genome bisulfite sequencing data (Figure 5C). The source data for Figure 5C are provided in Appendix A. Methylation level increased in the order CTCF < CGI < BG4 (median: ~10%, 15%, and 30%, respectively).

The total average BG4 peak methylation level and those of BG4 peak-overlapping fragments of CTCF and CGI were similar (median: 30%). Although both CGIs and CTCF peaks were hypomethylated, methylation was enhanced at their intersections (mostly BG4-harboring; median: 21%). To summarize, CTCF-bound sites and G4s have slightly higher methylation levels than CGIs in general. These results argue against the decisive role of hypomethylation maintenance in CTCF recruitment to CGIs and supports the role of local methylation.

### 2.3. G4s May Recruit HMG Proteins That Prevent Chromatin Condensation and CTCF Aggregation

Apart from sequence and methylation, CTCF recruitment depends on nucleosome arrangement [15,16]. We searched for nucleosome regulators among known G4 binders [18,19] and analyzed a group of chromatin architecture-modifying non-histone proteins with intrinsic affinity for bent DNA [27]—high mobility group (HMG) proteins—in a previously reported G4-interactome dataset [19].

We ranked the proteins based on their scores in the reported microarray-based assay with model G4s [19] and picked three hits(HMGN3, HMGN1, and HMGB2) for further analysis (Figure 6A). HMGN1 and HMGN3 are linker histone analogs. They bind to nucleosomal DNA and prevent internucleosome contacts [27,28] to facilitate chromatin decondensation, which renders linker DNA accessible to CTCF. HMGN3 also facilitates chromatin decondensation by recruiting histone acetyltransferases to nearby nucleosomes [29]. HMGB2 typically binds to linker DNA at a nucleosome entry point and acts as a CTCF insulator to prevent its abnormal aggregation [30].

To verify the association between the HMG proteins and G4s in the human genome, we examined available ChIP-seq data for HMGN3 (K562 cells), HMGN1 (CD4+ T cells), and HMGB2 (IMR90 cells). Average G4-seq coverage of protein-bound sites with flanks (±200 bp to account for the nearest nucleosome) was approximately 22%, 5%, and 4% for HMGN3, HMGN1, and HMGB2, respectively. These values were substantially higher than the average whole-genome G4-seq coverage (1.8%), indicating that G4s are enriched in HMG protein-bound sites. For HMGN3, frequent colocalization with G4s became even more apparent when we switched from G4-seq to BG4 peaks in K562 cells (Figure 6B): 77% of the 8940 BG4s overlapped with HMGN3 peaks and an additional 5% flanked HMGN3 peaks (±200 bp).

HMGB2 showed the least pronounced association with G4s, supposedly because it interacts non-specifically with any DNA kinks [30], and the resulting ‘noise signal’ partially overshadows specific binding. We assumed non-specific interactions to be minimal at low protein concentrations and analyzed senescent cells (IMR90p28) characterized by low HMGB2 expression levels [31]. G4-seq coverage of HMGB2-bound sites ± 200 bp in these cells (~11%) was almost 3-fold higher than in actively proliferating (IMR90p10) cells (4%), and 7-fold higher than in the whole genome (1.8%). This result suggests that HMGB2 tends to bind G4 sites in the first place.

To assess the significance of the association between G4s and HMG proteins, we first used X^2^ statistics. The observed G4-protein peak intersection frequencies exceeded those predicted by chance (*p* < 0.005), with the X^2^ statistic decreasing in the order HMGN3 >> HMGB2 (IMR90p10) > HMGN1 (Appendix A). Next, we performed permutation-based testing with Monte Carlo simulations. A comparison of randomized site intersections and real ones confirmed a non-random distribution in all cases (*p* < 0.001; Figure 6C). Finally, we repeated the permutation-based testing with additional filters applied to the input datasets to ensure that the results were qualitatively independent of the peak-calling procedure (Text box S1).

Given that G4s are associated with active transcription [32] and HMG proteins promote chromatin decondensation [27,28], it is possible that the whole-genome analysis of their relative distributions was biased. We therefore repeated the above experiments using available ATAC-seq data and focusing exclusively on open chromatin regions (Appendix A). In all cases, the comparison of randomized site intersections with real ones confirmed a non-random distribution (*p* < 0.00001), demonstrating that HMG proteins tend to bind G4 sites both at the whole-genome level and in open chromatin.

### 2.4. G4s Interact with HMG Proteins and Are Enriched in HMGN3- and CTCF-Bound Sites

The previously reported protoarray assay [19] allowed only semi-quantitative characterization of G4-protein binding; moreover, it was performed with random (model) G4s. We aimed to quantitatively characterize HMG protein binding with representative G4s from the protein occupancy sites and evaluate the binding selectivity. To identify representative G4s, we searched for enriched sequence motifs within HMGN3, HMGB2, and HMGN1-bound sites (Appendix A). In the case of HMGN3, motifs found in both G4-seq peak-intersecting and peak non-intersecting sites included G4-prone sequences or their complements and oligo-T/oligo-A–containing sequences. In the case of HMGB2, we found few motifs, so we also searched for specific sequence patterns in protein peak-intersecting G4 motifs in proliferating and senescent cells using scripts that were developed in-house (Appendix A). For HMGN1, the set of G4-seq peak non-intersecting sites was very large, so we analyzed only G4-seq peak-intersecting sites. The identified motifs included G/C- and A/T-rich and mixed sequences.

Several motif-matching ODNs (Appendix A) were obtained for each protein and analyzed by CD spectroscopy (Appendix A). The ODN sets included duplexes and G4s of various topologies, which enabled preliminary verification of HMG specificity for particular types of secondary structure. In the genomic context, all G4s are presumed to be intramolecular. As such, G4 ODNs that tended to form intermolecular structures (aggregates confirmed by PAGE) were excluded from further analysis (Appendix A). The CD data showed that most G4s had parallel or mixed topologies. We therefore obtained additional ODNs that reportedly adopt antiparallel-stranded quadruplex structures [22CTA [33], HRAS [34], and htel21T18 [35]. Each HMG protein was also tested for binding with non-structured ssDNA (A20 and T20), the model hairpin ds26 [36], and parallel G4 from the microarray assay [19] (positive control). That control G4 has been referred to as G4-2 [19] and CT1 [37] in previous works.

ODN interactions with HMG proteins were analyzed by MST; binding curves and Kd values are shown in Figure 7A and Table 1. Parallel and hybrid G4s bound with HMG proteins in the high nanomolar/low micromolar concentration range. In contrast, antiparallel G4s, motif-matching duplexes, hairpins, and non-structured ODNs showed low or no affinity for HMG proteins. These results explain the partial colocalization of the protein occupancy sites with G4-seq peaks. To summarize, we showed that parallel-stranded and mixed-topology G4s bind with the HMG proteins in vitro and may recruit them in the genome.

To verify whether HMG protein recruitment to G4 sites contributes to CTCF binding, we compared relative distributions of HMG and CTCF ChIP-seq peaks. Overall, there was low to moderate colocalization: CTCF peaks overlapped with 53% of HMGN3 peaks, 16% of HMGB2 peaks, and 8% of HMGN1 peaks in the respective cell lines. Substantial portions of the colocalized sites (30% of HMGN3 peaks, 28% of HMGB2 peaks, and 31% of HMGN1 peaks) harbored G4-seq peaks, although overall frequency of triple intersections was low (Figure 7B). Statistical analysis confirmed significance of G4 contribution to the association between HMGN3 and CTCF [X^2^(1, N = 37690) = 5574.1, *p* < 0.01]. Switching from G4-seq to G4-ChIP-seq (BG4) peaks increased the frequency of triple intersections (HNGN3-G4-CTCF) in K562 cells. Those triple intersections accounted for 60% of BG4 peaks. We conclude that HMGN3 recruitment to G4 sites may partially contribute to subsequent CTCF positioning.

## 3. Discussion

We showed that CTCF binds folded G4s and consensus duplexes with comparable affinities (Figure 4). This is hardly surprising, because G4 recognition has been reported previously for other zinc-finger transcription factors with G-rich binding motifs, such as Sp1 [38,39] and MAZ [40]. The latter is functionally rather similar to CTCF, and the two work together as insulators to shape topologically associating domains (TAD) and sub-TAD domains. However, CTCF-G4 and MAZ-G4 complexes are unlikely to exist throughout the G0/G1 phase, except in the cases of persistent (e.g., ligand-stabilized) G4s. The majority of G4s are transient and observed upon replication [41]. Thus, we argue that direct contacts only partially account for CTCF recruitment to G4-prone sites. Histone marks and linker histone analogs that render linker DNA accessible to CTCF may be as important as sequence. We showed that three chromatin modulators (HMGN3, HMGB2, and HMGN1) recognize G4 structures in a topology-specific manner, and the recruitment of HMGN3 to G4 sites may contribute to subsequent CTCF positioning at those sites (Figure 7). As concerns CpG methylation, we argue that global G4-dependent DNMT1 inhibition in CGIs [4] is not crucial for CTCF positioning (Figure 5). However, we do not question the importance of local methylation due to its well-established effects on DNA affinity for zinc fingers [10] or stability of DNA secondary structures [4]. We conclude that a multistep mechanism involving G4-dependent modulation of nucleosome density, positioning, and internucleosome contacts [18,19] and DNA secondary structure-specific CTCF binding may be at play in the genome.

Our findings support G4 involvement in chromatin organization and partially explain G4 enrichment at TAD boundaries [6]. These findings also add to the growing body of links between G4s and the epigenetic machinery, highlighting the prospects of G4s as epigenetic drug targets and raising new concerns about possible side effects of G4-stabilizing small molecule therapeutics [2,42]. Moreover, evidence for G4-CTCF binding adds complexity to the current vision of enhancer-promoter interactions at G4-prone sites [6,43]. One example of such complexity is the MYC case. It is also of particular interest with respect to anticancer strategies based on 3D genome reorganization [44]. In several cancer cell lines, the CTCF binding site upstream of the MYC promoter accounts for long-distance interactions with cancer-specific downstream super-enhancers, which result in elevated MYC expression [45]. In the majority of cell lines, CTCF clustering at the promoter boundary represses MYC transcription, i.e., the CTCF-induced insulation ensures basal rather than elevated expression [46,47]. In K562 cells, we were unable to alter CTCF occupancy in the MYC promoter with G4 ligands, supposedly because G4s (and the cluster of CTCF-bound sites) were already present in the absence of the ligands (Figure 3). It is also possible that PDS failed to stabilize MYC G4, even though it stabilizes the homologous G4 structure Pu22 [22]. The G4-prone site in STAT3 gene that did not coincide with a high confidence BG4 peak and was the most responsive to ligand treatment (Figure 3). Thus, persistent G4s appear to be more challenging than transient ones in terms of fine-tuning with exogenous compounds. We hope that our results along with other emerging evidence for G4 interference with 3D genome organization [42] will stimulate studies of G4-targeting agents as epigenetic drug candidates.

## 4. Materials and Methods

### 4.1. Cell Culture Treatment with G4 Ligands and Chromatin Immunoprecipitation

K562 cells were cultured in RMPI-1640 medium (Paneco, Moscow, Russia) supplemented with 0.4% fetal bovine serum (HyClone GE Healthcare, Greater Milwaukee Area, WI, USA). Cell viability was quantified via trypan blue staining. The G4-stablizing ligands (PDS or PhenDC3) were added to the cell suspensions (1–1.25 mln/mL) to final concentrations of 10 or 20 μM (PDS) and 8 μM (Phen DC3). After 24 h (10 μM PDS) or 48 h (20 μM PDS or 8 μM PhenDC3) of incubation, cells were fixed, and chromatin immunoprecipitation (ChIP) was performed using SimpleChIP^®^ Plus Enzymatic Chromatin IP Kit with magnetic beads (Cell Signalling Technology, Danvers, MA, USA) following the manufacturer’s protocole. Briefly, crosslinking was performed via formaldehyde treatment and quenched with glycine. Cells were harvested and nuclei were prepared by incubation with the lysis buffer. Chromatin digestion was performed by treatment with micrococcal nuclease and stopped by EDTA treatment. Next, the lysate was sonicated to obtain DNA fragments of 200–500 bp. Per chromatin immunoprecipitation (ChIP) reaction, ∼5–10 μg of digested, cross-linked chromatin was incubated with 2–4 μg CTCF antibody pAb (Active Motif, Carlsbad, CA, USA) overnight at 4 °C. Normal Rabbit IgG (Cell Signaling Technology, Danvers, MA, USA) was used as a negative ChIP control. On the next day, Protein G Magnetic Beads (Cell Signaling Technology, Danvers, MA, USA) were added in each sample and incubated for 6 h at 4 °C. Immobilized complexes were washed two times for 10 min at 4 °C in low salt (1X ChIP buffer) and high salt (1× ChIP buffer supplemented with 350 mM NaCl) solutions. Samples were incubated with RNase A (Cell Signaling Technology, Danvers, MA, USA) in TE buffer (10 mM Tris-HCl, pH 8.0, 1 mM EDTA) for 30 min at room temperature. The DNA was eluted from the beads and decrosslinked by proteinase K digestion for 2 h at 65 °C. Next, the DNA was purified using DNA purification spin columns and analyzed by Illumina Next Generation sequencing or qRCR.

### 4.2. ChIP-Seq and Data Analysis

Paired-end libraries were prepared according to the manufacturer’s recommendations using NEBNext Ultra II DNA Library Prep Kit (New England Biolabs, Ipswich, MA, USA). The libraries were indexed with NEBNext Multiplex Oligos kit for Illumina (96 Index Primers, New England Biolabs, Ipswich, MA, USA). Size distribution for the libraries and their quality were assessed by Agilent Bioanalyzer using Agilent DNA High Sensitivity Chips (Agilent Technologies, Santa Clara, CA, USA). The libraries were subsequently quantified by Quant-iT DNA Assay Kit, High Sensitivity (Thermo Scientific, Waltham, MA, USA). DNA sequencing was performed on the HiSeq 2500 platform (Illumina, Madison, WI, USA) according to the manufacturer’s recommendations, using the following reagent kits: HiSeq Rapid PE Cluster Kit v2, HiSeq Rapid SBS Kit v2 (200 cycles), HiSeq Rapid PE FlowCell v2 and a 1% PhiX spike-in control. The experiment was repeated twice. Reads for each biological replicate were mapped to the human genome (version hg19) using Bowtie2 (version 2.2.3) with the ‘—very-sensitive’ preset [48]. Non-uniquely mapped reads, PCR duplicates and reads with MAPQ < 30 were filtered out with ‘samtools rmdup’ and ‘samtools view -h -F 256 -q 30’. Peaks were called using PePr (https://github.com/shawnzhangyx/PePr, accessed on 20 December 2020) with a *p*-value cutoff of 0.05 and a sliding window size of 100 bp [49]. The bigWig files were generated using deepTools, version 2.0 [50].

### 4.3. ChIP-qPCR

For qPCR analysis of the immunoprecipitated samples and inputs, primers intersecting or flanking the CTCF occupancy sites of interest (those intersecting BG4 peaks, G4-seq peaks, G4 motifs or none of the above) were designed using NCBI primer-BLAST and Eurofins Genomics primer design tools (Appendix A). qPCR experiments were performed by QuantStudio5 (Thermofisher Scientific, Waltham, MA, USA) using 96-well white plates. In each experiment, immunoprecipitated samples, primers, and SYBR Green PCR Master Mix—a certified solution of a hot-start DNA polymerase, dNTPs, MgCl_2_, enhancers, and stabilizers (GenTerra, Russia)—were mixed to give a final volume of 25 μL. The thermocycler program was 95 °C for 5 min (1 cycle) followed by a 2-step reaction of 95 °C for 10 s and 60 °C (for primer pairs STAT3 and VEGFA) or 62 °C (for primer pairs MYC, USP24 and VPS4A) for 30 s (40 cycles). Amplification curves were analyzed using Thermofisher software and ΔΔCq values were calculated in Microsoft Excel to express the relative DNA amplification of the CTCF-occupied region in ligand-treated/non-treated samples to inputs:

ΔΔCq = 2^(−ΔCq_sample)/2^(−ΔCq_input);

ΔCq = Cq_region of interest—Cq_control region

The control region was a G4-lacking site USP24. All primers are specified in Appendix A. All experiments were performed in two biological and two technical replicates.

### 4.4. Molecular Modeling

For G4 docking to CTCF, we used the available model of CTCF zinc fingers 4–10 [10] in complex with a DNA duplex (PDB ID: 5UND) and removed the duplex. The MYC-G4 model, referred to as ‘Pu27 truncated’ in a previous work (29) was obtained from the Pu24T model (PDB ID: 2N6C) as described previously [51]. A two-step docking procedure was carried out following a published protocol [52]. Briefly, the 1st step included ‘rigid’ docking using Hex 8.0.0. software [52] and post-processing MM minimization using OPLS force field. At step 2, complexes selected based on the scoring function during post-processing at step 1 were minimized using SYBYL software (Certara, Princeton, NJ, USA) and Powell method. Parameters for interatomic interactions and partial charges on the atoms were taken from Amber7ff02 force field.

### 4.5. Bioinformatics

G4 motif and i-motif mining was performed using the updated version of imGQFinder [37] and G4/iM-Grinder [53]. G4-seq and G4-ChIP-seq (BG4) data were downloaded from Gene Expression Omnibus. ChIP-seq data for CTCF and other proteins and ATAC-seq data for respective cell lines were downloaded from ENCODE or Gene Expression Omnibus. Data on CpG methylation were obtained from ENCODE. Intersections, evaluation of relative distances, and other manipulations with the sets of genomic intervals were performed using Bedtools [54]. Statistical significance of the intersections between G4 sites and protein-bound sites was verified by chi-squared tests, Fisher tests and permutation-based tests. Enriched motifs in the protein-bound peaks and individual matching sequences were identified using MEME [55] and FIMO [56] algorithms. Details on all bioinformatics analyses are given in Appendix A.

### 4.6. Oligonucleotides, Recombinant Proteins, and Small-Molecule Ligands

Oligonucleotides (ODNs) were obtained from Litekh, Moscow, Russia (purity ≥ 95%, HPLC). Recombinant mouse CTCF with polyhistidine (His) tag was purchased from MyBioSource, San Diego, CA, USA. Recombinant human HMGB2 with His tag was purchased from Abcam, Cambridge, UK. Recombinant human HMGN1 and HMGN3 with His tags were purchased from LifeSpan BioSciences, Seattle, WA, USA. For microscale therophoresis (MST) assays, the proteins were labeled with the His-Tag Labeling Kit RED-tris-NTA (NanoTemper Technologies, Munich, Germany) according to the manufacturer’s protocol. G4 ligands pyridostatin (PDS) and bisquinolinium-derivatized phenanthroline-dicarboxamide (Phen DC3) were obtained from Sigma-Aldrich (St. Louis, MO, USA).

### 4.7. Circular Dichroism Spectroscopy and Electrophoresis

ODN solutions (40 μM) in 140 mM potassium phosphate buffer (pH 7.2 or 6.7) containing 10 mM NaCl were heated to 95 °C for five min and ice-cooled on ice (for G4s and i-motifs) or cooled slowly to room temperature (for duplexes) to facilitate correct folding. These annealed samples were used as stock solutions for microscale thermophoresis assays. For circular dichroism (CD) measurements and polyacrylamide gel electrophoresis (PAGE), the pre-annealed samples were diluted with the respective buffers to a final concentration of 1–5 μM. CD spectra were recorded at room temperature using a Chirascan spectrophotometer (Applied Photophysics, Leatherhead, UK) and a quartz cuvette with an optical path length of 10 mm. Nondenaturing polyacrylamide gel electrophoresis (PAGE) was performed in a standard Tris–borate–EDTA (TBE) buffer (pH 8) at a gel concentration of 20%. Low molecular weight marker 10–100-nt ssDNA (Affymetrix, Santa Clara, CA, USA) was used as a control. The gels were run for 2 h at 200 V at room temperature with a 1× TBE with 10 mM KCl buffer. Oligonucleotide bands were stained with SYBR Gold (Thermo Fisher Scientific, Waltham, MA, USA) and visualized using a Gel Doc scanner (Bio-Rad, Hercules, CA, USA).

### 4.8. Microscale Thermophoresis

Labeled proteins were mixed with two-fold serial dilutions of unlabeled ODNs to a final protein concentration of 50 nM and varying ODN concentrations (from 20 µM to 0.61 nM). The mixtures were stored at room temperature for 15 min prior to MST measurements. MST curves were registered using Monolith NT.115, equipped with a RED/GREEN detector, and standard capillaries (NanoTemper, Munich, Germany) at 22 °C with MST monitoring by fluorescence of the labeled protein. The dependence of its normalized fluorescence on the concentration of the unlabeled oligonucleotide was analyzed using MO.Affinity Analysis software (NanoTemper, Munich, Germany). To obtain dissociation constant values, experimental data were fitted to the Kd model.

## Figures and Tables

**Figure 1 ijms-22-07090-f001:**
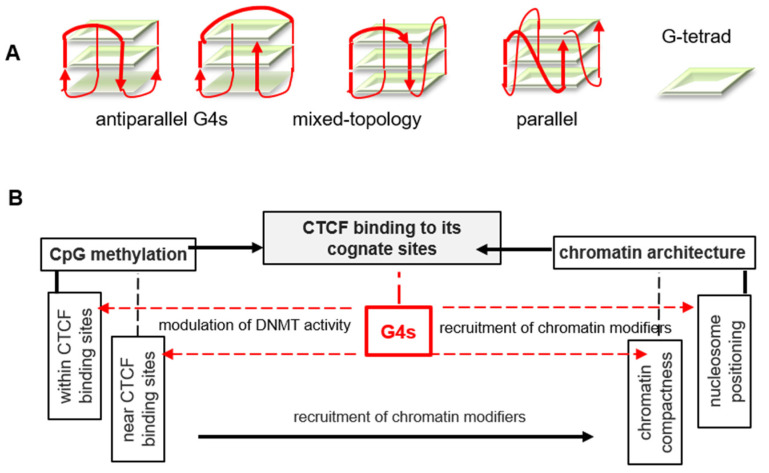
G-quadruplex (G4) structures and their presumed role in CTCF positioning. (**A**) Schematic representation of G4s. (**B**) Summary of the major factors that affects CTCF binding to genomic DNA. Dashed red lines indicate presumed links that are verified in this study.

**Figure 2 ijms-22-07090-f002:**
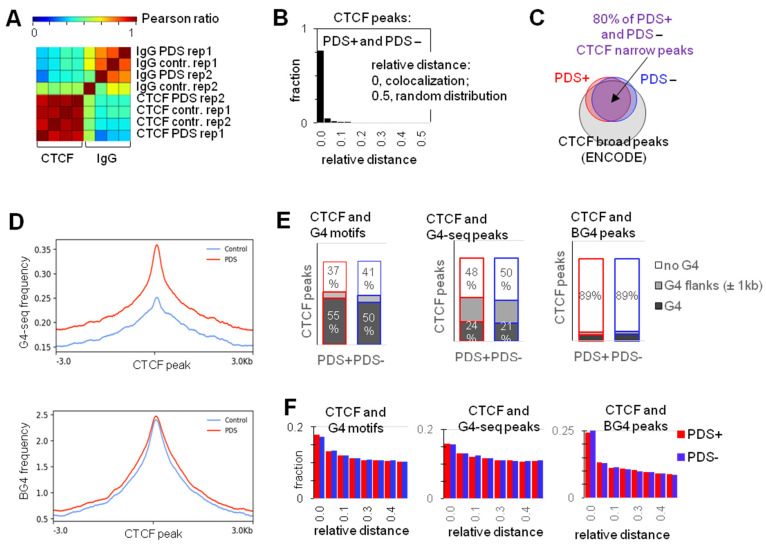
CTCF positioning relative to G4 sites in pyridostatin (PDS)-treated and non-treated cells. (**A**) Summary of the CTCF ChIP-seq data: Pearson correlation between ChIP-seq data sets obtained for PDS-treated (PDS+) and non-treated (PDS−) K562 cells (two biological repeats). (**B**) Relative distance between PDS+ and PDS- CTCF ChIP-seq peaks. (**C**) Venn diagram summarizing the overlap between PDS+/− CTCF peaks and the previously reported CTCF ChIP-seq peaks. (**D**) Metaplots illustrating G4-seq and BG4 peak distribution relative to PDS+ or PDS-(control) CTCF peaks. (**E**) Summary of CTCF peak intersections with G4 sites. (**F**) Relative distances between CTCF peaks and G4 sites.

**Figure 3 ijms-22-07090-f003:**
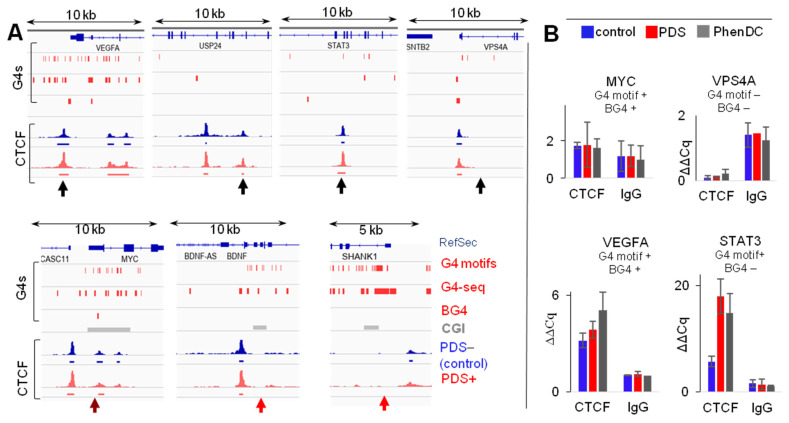
CTCF occupancy at G4 sites in PDS-treated and non-treated cells. (**A**) Genome browser snapshots for VEGFA, USP24, STAT3, VPS4A, MYC, BDNF and SHANK promoter regions/gene bodies harboring G4 motifs, G4-seq peaks, G4 ChIP-seq (BG4) peaks and/or CTCF-bound sites in pyridostatin (PDS)-treated and non-treated K562 cells. Black arrows mark sites analyzed in ChIP-qPCR assays. The CTCF-positive G4-negative site in USP24 was used as a reference, and the CTCF-negative G4-negative site in VPS4A was used as a negative control. Red arrows mark sites predicted to form stable G4s and i-motifs that were used in the binding assays. (**B**) ChIP-qPCR results. The histograms illustrate normalized CTCF occupancy at G4-prone (MYC, VEGFA, and STAT3) and non-G4 (VPS4A) sites relative to the control site (USP24) in non-treated (blue) K562 cells and those treated with PDS (red) or PhenDC3 (grey).

**Figure 4 ijms-22-07090-f004:**
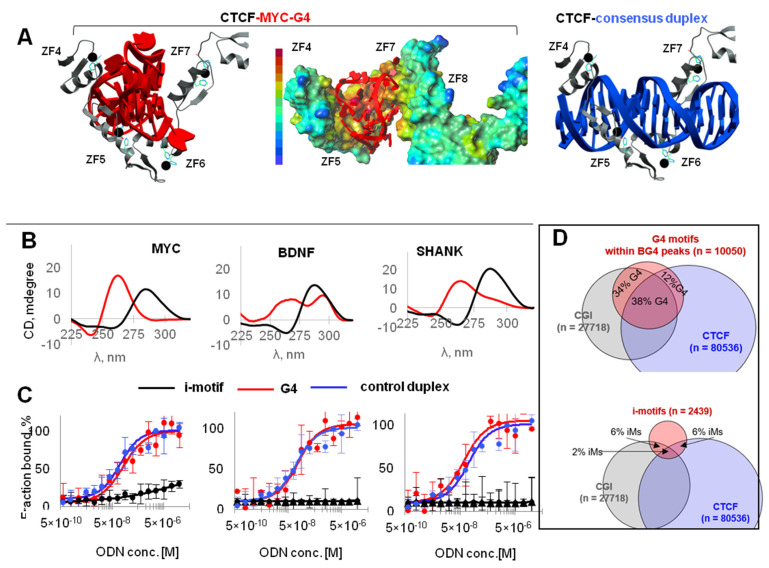
Analysis of CTCF-G4 interactions. (**A**) In silico verification of G4-CTCF binding. The best binding energy conformation of the complex obtained by docking MYC-G4 (red) to CTCF is shown. The complex with the consensus duplex (blue) is shown for comparison. In the middle panel, the surface of MYC-G4-bound CTCF is colored according to the electrostatic potential: from negative (blue) to positive (red). (**B**) Circular dichroism spectra of the 2 µM G4 (red) and iM (black) solutions in 140 mM potassium-phosphate buffer, pH 6.7, supplemented with 10 mM NaCl, obtained at 22 °C. (**C**) Microscale thermophoresis (MST)-based analysis of CTCF interactions the G4s (red), respective i-motifs (black) and the control duplexes (blue). (**D**) Comparison of G4 motif and i-motif intersections with CTCF-bound sites in K562 cells.

**Figure 5 ijms-22-07090-f005:**
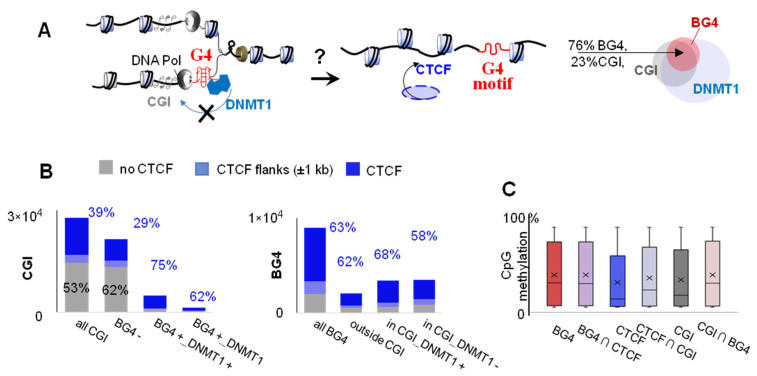
Verification of G4 contribution to the link between CpG islands (CGIs) and CTCF recruitment. (**A**) Hypothetical schemes summarizing the link between G4s, CTCF and CGI methylation: G4s are enriched in CGIs and supposedly protect CGIs from methylation by inhibiting DNMT1, which is favorable for CTCF binding. (**B**) Intersections of CTCF peaks with various subsets of CGIs and BG4 peaks. (**C**) Analysis of average CpG methylation levels in CpG-containing BG4 peaks, CTCF peaks, CGIs and their intersections in K562 cells.

**Figure 6 ijms-22-07090-f006:**
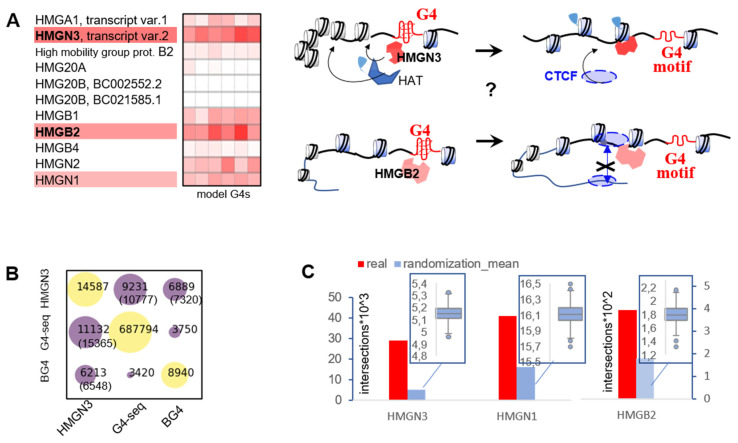
Search for G4-binding chromatin modulators that may affects CTCF positioning. (**A**) A heatmap summarizing G4 binding affinity (while, low; red, high) of HMG proteins in previous assays and hypothetical schemes summarizing the link between G4s, HMG and CTCF: the presumed recruitment of HMGN1 and HMGN3 to G4-prone sites induces chromatin decondensation by the disruption of internucleosome contacts and activation of histone acetyltransferase (HAT). Recruitment of HMGB2 prevents CTCF aggregation. (**B**) Intersection table for HMGN3 ChIP-seq, G4-seq and G4 ChIP-seq (BG4) peaks in K562 cells. The values equal to the number of intersections. Data in parentheses represent the number of G4-seq/BG4 peaks intersecting HMGN3 peaks with flanks (±200 bp). (**C**) Whole-genome analysis of the intersections between G4-seq peaks and HMG protein-bound sites and significance of the G4-HMG correlations (Monte-Carlo simulation results). Red boxes indicate the true number of G4-seq-overlapping HMG protein peaks. The boxplots refer to the randomized peaks.

**Figure 7 ijms-22-07090-f007:**
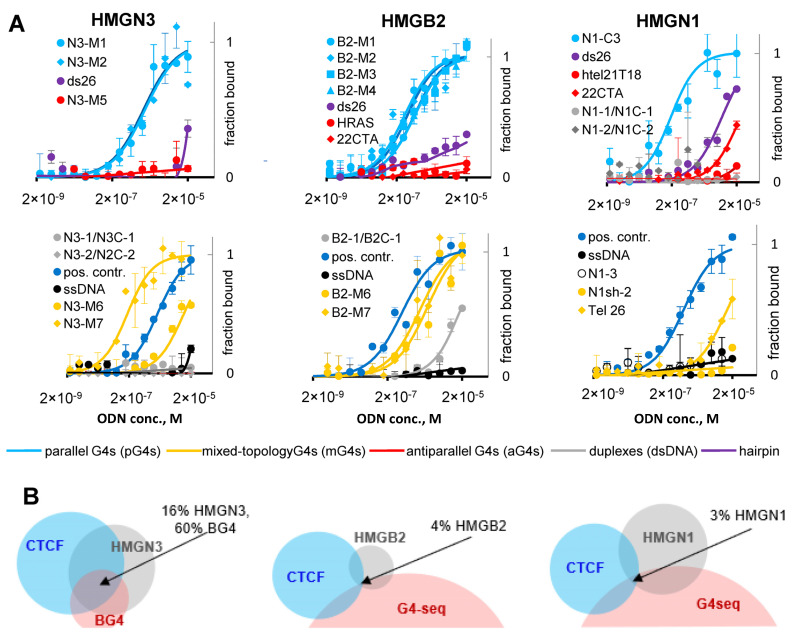
G4-HMG protein interactions and their possible contributions to CTCF positioning. (**A**) Microscale thermophoresis (MST)-based analysis of HMG proteins interactions with representative sequences that comply with enriched motifs (G4s and duplexes), the control hairpin and single-stranded oligonucleotides. Conditions: 50 nM labeled protein, 0–20 µM oligonucleotide, 140 mM potassium-phosphate buffer, pH 7.2 (6.7 for the i-motif N1-3). (**B**) Venn diagrams illustrating the overlaps between G4-ChIP-seq (BG4) peaks, HMGN3-bound sites, and CTCF-bound sites in K562 (left); G4-seq peaks, HMGB2-bound sites, and CTCF-bound sites in IMR-90; and G4-seq peaks, HMGN1-bound sites, and CTCF-bound sites in CD4+ T-cells.

**Table 1 ijms-22-07090-t001:** Summary of HMG protein interactions with G4s and control oligonucleotides.

ODN	Kd, µM (HMGN3)	Kd, µM (HMGB2)	Kd, µM (HMGN1)
pG4s	1.5 ± 0.6	0.6 ± 0.3	0.2 ± 0.1 (N1–C3); 1.6 ± 0.3 (pos. contr.)
aG4s	>>20	>>20	≥20
mG4s	≥10 (M6); 0.15 ± 0.06 (M7)	4 ± 2 (M6); 1.6 ± 0.6 (M7)	≥10 (Tel 26); >20 (N1sh-2)
hairpin	≥20	≥20	≥10
dsDNA	>>20	>>20	>>20
ssDNA	>>20	>>20	>>20

## Data Availability

CTCF ChIP-seq data for PDS-treated and non-treated K562 cells have been deposited with Gene Expression Omnibus under accession number GSE173074. The updated version of ImGQFinder is an open source collaborative initiative available in the GitHub repository (https://github.com/RCPCM-GCB/ImGQFinder, accessed on 15 November 2020).

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
