# Peer review of "DNA G-Quadruplexes Contribute to CTCF Recruitment"

_ijms, 2021, doi:10.3390/ijms22137090_

Round 1
Reviewer 1 Report
The manuscript by Tikhonova et al is thorough and well written. With minor reservations, as detailed below, this reviewer recommends accepting the manuscript.
Minor weakness: Pyridostatin is not as "pan" of a G4-stabilizer as other agents, such as TMPyP4 or BRACO-19. It is not actually a stabilizer of the MYC structure, and thus the conclusions drawn from Figure3A bottom needs to be revised. The lack of a difference in CTCF occupancy of the MYC G4 region in the presence of PdSA is likely due to a lack of effect of the molecule on the G4.
Author Response
We are very thankful for the reviewer’s remarks, agree on the main issue and have corrected the manuscript text accordingly. The following comments have been added on page 3 (the last paragraph) and page 11 (discussion, paragraph 2):
“Both PDS and Phen-DC3 are presumed to be pan-quadruplex ligands [23,24]. However, it should be noted that their effects on conformationally polymorphic G4-prone sequences, such as MYC NHE [25], require further investigation”.
“It is also possible that PDS failed to stabilize MYC G4, even though it stabilizes the homologous structure Pu22.”
We now see that PDS-induced stabilization of MYC G4 is indeed questionable. Several groups have reported PDS to enhance the stability of Pu22 (a mutant fragment of Pu27), but the effects on the wild-type full-length Pu27 await thorough evaluation. This would be a is challenging task, because Pu27 adopts various structures, including the intermolecular ones, in vitro, which complicates melting assays. Despite the conformational polymorphism of MYC G4, we decided to include this quadruplex in our analysis because of its biological significance. Nevertheless, we admit that future bona fide verification of MYC G4 stability in the presence/absence of the ligands is needed.
TmPyP4 and BRACO 19 are hardly superior to PDS and PhenDC3 with respect to our assays. TmPyP4 may destabilize G4s in some cases [Lejault et al. Cell Chemical Biology, Cell Press, 2021, 28 (4): 436 - 455]. BRACO-19 has insufficient quadruplex-over-duplex selectivity [Machireddy et al. Molecules. 2019, 24(6): 1010; Read et al. PNAS 2001, 98(9):4844-9].
Reviewer 2 Report
The manuscript might be acceptable, but either I don't have access to the
Supplementary materials or these files are missing, so without them
I can't give a clear answer as to whether the manuscript is suitable
for publication.
Author Response
We thank the reviewer and apologize for the inconvieniece. We must have lost the file upon submission. Please see the attached pdf.

Round 2
Reviewer 2 Report
The submitted manuscript with supplemental materials is a successful work of the authors, in terms of content and graphics. The manuscript contains a lot of scientifically valuable information. Some parts are narrowly specialized, and their understanding is more difficult for readers which are not working in the studied field. The authors have used a wide range of experimental methods in their experiments to validate obtained results; they enriched the knowledge concerning a direct and indirect role of G4s in CTCF recruitment. I have not found serious flaws; therefore, I recommend the manuscript to publish as it.